# Synthesis of Nitrogen-Doped Graphene on Copper Nanowires for Efficient Thermal Conductivity and Stability by Using Conventional Thermal Chemical Vapor Deposition

**DOI:** 10.3390/nano9070984

**Published:** 2019-07-07

**Authors:** Minjeong Park, Seul-Ki Ahn, Sookhyun Hwang, Seongjun Park, Seonpil Kim, Minhyon Jeon

**Affiliations:** 1Department of Nanoscience and Engineering, Center for Nano Manufacturing, Inje University, Gimhae 50834, Korea; 2Department of Military Information Science, Gyeongju University, Gyeongju 38065, Korea

**Keywords:** copper nanowires, N-doped graphene, chemical vapor deposition, growth, double-zone growth process, thermal interface materials

## Abstract

Cu nanowires (NWs) possess remarkable potential a slow-cost heat transfer material in modern electronic devices. However, Cu NWs with high aspect ratios undergo surface oxidation, resulting in performance degradation. A growth temperature of approximately <1000 °C is required for preventing the changing of Cu NW morphology by the melting of Cu NWs at over 1000 °C. In addition, nitrogen (N)-doped carbon materials coated on Cu NWs need the formation hindrance of oxides and high thermal conductivity of Cu NWs. Therefore, we investigated the N-doped graphene-coated Cu NWs (NG/Cu NWs) to enhance both the thermal conductivity and oxidation stability of Cu NWs. The Cu NWs were synthesized through an aqueous method, and ethylenediamine with an amine group induced the isotropic growth of Cu to produce Cu NWs. At that time, the amine group could be used as a growth source for the N-doped graphene on Cu NWs. To grow an N-doped graphene without changing the morphology of Cu NWs, we report a double-zone growth process at a low growth temperature of approximately 600 °C. Thermal-interface material measurements were conducted on the NG/Cu NWs to confirm their applicability as heat transfer materials. Our results show that the synthesis technology of N-doped graphene on Cu NWs could promote future research and applications of thermal interface materials in air-stable flexible electronic devices.

## 1. Introduction

The performance enhancement and long-term reliability of microelectronic devices are decidedly key issues in a wide range of electronic applications. Non-efficient heat dissipation in such devices leads to many problems, such as malfunction, decreased performance, and short-term stability as the power densities are increased. Therefore, effective heat transfer from integrated circuit to heat sink should contribute to the development of modern electronic devices [1,2,3,4,5,6,7,8].

Recently, various nanostructures, such as nanowires (NWs) [9,10,11,12,13] and nanoparticles [14,15,16,17], have been successfully synthesized as heat-transfer materials. Among these, NWs are more suitable as heat-transfer materials, owing to their natural continuity as well as high aspect ratio, which could lead to lower percolation thresholds. Their characteristics, such as wire diameter, density of wires in the network, and junction resistance substantially affect the physical properties [18,19]. Conventional metal NWs, especially Cu NWs, have a number of advantages, such as low cost, abundance on Earth, and has a similar electrical conductivity to silver. Therefore, NWs possess remarkable potential as low-cost heat-transfer materials in modern electronic devices. However, Cu NWs with high aspect ratios could be intrinsically unstable under ambient conditions. The rapid surface oxidation of Cu NWs weakens their physical properties [20,21]. Several efforts have been made to prevent the surface oxidation of Cu NWs [22,23,24,25,26,27,28,29,30]. One approach is to provide a protection layer using graphene [28,29,30]. Carbon materials, such as carbon nanotubes and graphene, have been widely studied as heat-transfer materials based on their high intrinsic thermal conductivities, which range from 3000–6000 W/m∙K [8]. Here, to realize effective heat transfer materials, various types of graphene are needed. The high thermal conductivity and formation hindrance of oxides on Cu NWs are also of great technological importance. Doping Cu NWs with other elements is a promising way to achieve this goal. Especially, nitrogen (N)-doped graphene promises many fascinating properties and widespread potential applications, such as super conduction [31], ferromagnetism [32], and has high thermal conductivity and the formation hindrance of oxides, such as CuO, Cu_2_O, and Cu(OH)_2_. Therefore, intensive theoretic studies are focused on N-doped graphene, and many theoretic models of the substitutionary-doped graphene have been established [33,34,35,36,37].However, these studies are only based on theory, and an experimental example of the substitutionary N-doped graphene is still lacking, attributed to the limitations of synthetic methods. For decades, chemical vapor deposition (CVD) has been used as the general method for growing graphitic or doped graphitic [33,34,35,36,37,38] thin films. However, growing graphene directly onto Cu NWs is very difficult because a conventional thermal CVD (T-CVD) process requires a high process temperature of approximately 1000 °C.

In this study, we demonstrated N-doped graphene-coated Cu NWs (NG/Cu NWs) as the heat-transfer materials synthesized using a conventional T-CVD. To achieve a low temperature growth by using a conventional T-CVD, we used the double-zone growth process (DZGP) and succeeded in growing graphene on Cu NWs at a low temperature. Here, the Cu NWs with an amine group was used for the N-doped graphene. We realized the synthesis of the substitutionary NG/Cu NWs. Moreover, the experimental measurements of the morphology and oxidation stability properties of the NG/Cu NWs are provided in this study, and an efficient thermal conductivity of NG/Cu NWs was indicated using the laser flash method. In this way, the applications of the thermal interface materials (TIMs) and air-stable flexible electronic devices can be largely improved and expanded.

## 2. Materials and Methods 

### 2.1. Materials

All the following chemicals were used as received without further purification: Sodium hydroxide was purchased from Alfa Aesar (35,633). Copper nitrate (Alfa Aesar, Haverhill, MA, USA, 10,699) was used as the copper precursor, ethylenediamine (EDA; Sigma Aldrich, St. Louis, MO, USA, E1521) was used as the structure-directing agent, and hydrazine (35 wt%, Sigma Aldrich, St. Louis, MO, USA, 309,400) was used as the reducing agent. The SiO_2_/Si wafer was purchased from LG Siltron Inc. Gumi, South Korea. (Silicon wafer; P/<100>, Boron) and was used as the substrate when we grew the N-doped graphene. The silicone elastomer (PDMS; Dow corning^®^, Midland, MI, USA, Sylgard 184 base & curing agent) was used as the polymer matrix when we fabricated TIMs.

### 2.2. Preparation of Cu NWs

The Cu NWs with diameters of 160–200 nm and length of ~25 µm were produced through an aqueous route according to a redox reaction [10,11]. In this study, 200 mL of sodium hydroxide solution (5 M), 10 mL of copper nitrate solution (0.1 M), and 1 mL EDA were added to a glass reactor, followed by thorough mixing of all reagents for 3 min. The reactor was placed in a water bath (60 °C), and then 500 µL of hydrazine was added into the reactor. After 1 h, the synthesis of Cu NWs was completed, and the Cu NWs were separated from the reaction solution by using a vacuum filtration system, in which a polyethersulfone (PES) membrane filter paper (pore size: 0.45 µm) was used. The Cu NWs/PES membrane filter paper was dried at 80°C in air. Here, the EDA with an amine group induces anisotropic growth of Cu, and can produce Cu wires. Thus, the Cu NWs with an amine group could be easily obtained from PES membrane filter paper.

### 2.3. Synthesis of the NG/Cu NWs

We used a T-CVD for fabricating NG/Cu NWs. Figure 1 shows the synthesis process of the NG/Cu NWs. Figure 1a shows the growth temperature of double zone, and the growth mechanism of the N-doped graphene in the quartz of the T-CVD. As shown Figure 1b, the as-grown Cu NWs were placed in the T-CVD quartz tube, which was heated up to 350 °C, to remove impurities, for 15 min. Here, we employed the DZGP in which the reactants heated at 900 °C in the first zone, flowed into the second zone and were maintained at 600 °C for the growth of N-doped graphene at a low temperature. In addition, the various partial pressures of CH_4_ to maintain the morphology of Cu NWs with amine group were systematically examined. Next, we introduced the reaction gas mixture composed of Ar, H_2_, and CH_4_ (4:1:1 and 10:2.5:1) into the quartz tube for 10 min. Here, the N-doped graphene was synthesized using the different carbon source (CH_4_) conditions (CH_4_ flow; 20 and 50 sccm). After the cooling step, we could easily obtain the NG/Cu NWs.

In addition, the heat-transfer materials were fabricated using as-grown Cu NWs, NG/Cu NWs synthesized with 20 sccm CH_4_ (NG/Cu NWs;M20), and NG/Cu NWs synthesized with 50 sccm CH_4_ (NG/Cu NWs;M50) for thermal conductivity measurements. The three prepared materials, epoxy matrix, and PDMS curing agent were mixed using a paste mixer for 10 min, and the pastes were then placed in oven at 120–180°C for 30 min. Finally, we obtained three heat-transfer materials.

### 2.4. Characterization

The sheet resistances of as-grown Cu NWs, NG/Cu NWs;M20, and NG/Cu NWs;M50 were measured using a 4-point probe system (DASOL ENG, Chungcheongbuk-do, South Korea, FPP-HS 8). Raman spectra of NG/Cu NWs;M20 and NG/Cu NWs;M50 were obtained using a 532 nm excitation laser with spot diameter of 50 μm (JASCO, Tokyo, Japan, NRS-3300). An elemental composition analysis was conducted using X-ray photoelectron spectroscopy (XPS, Thermal Fisher Scientific, Waltham, MA, USA, ESCALAB 250 XPS system, a monochromatic X-ray source, image resolution <3 μm). Morphological properties of the three materials were investigated using a field emission scanning electron microscope (FE-SEM, Hitachi, Tokyo, Japan, S-4300), and the thermal conductivities were measured using a laser flash method (NETZSCH, Selb, Germany, LFA467, Temperature range from −100°C to 1250°C, Xenon flash lamp; pulse energy: Up to 10 J/pulse, pulse width: 10 to 1500 μs).

## 3. Results and Discussion

In this work, an aqueous procedure was proposed to synthesize large-scale Cu NWs. We obtained approximately 53.3 mg of Cu NWs from 64.04 mg of Cu solution at one time. This result shows approximately 83.2% yield of Cu NWs. FE-SEM was used to observe the morphology of as-grown Cu NWs, as shown in Figure 2. The average diameter of the Cu NWs immediately after growth was about 400 nm and the average length was ~25 µm. Regrettably, the Cu NWs with high aspect ratio are easily oxidized in air [28,29,30], and the physical properties of Cu NWs are weakened by oxidation [20]. Therefore, we synthesized NG/Cu NWs via DZGP to inhibit oxidation. The average diameter of Cu NWs after graphene growth was reduced by the growth heat and the Cu NWs immediately after growth were about 200 nm. To evaluate the morphologies of NG/Cu NWs;M20 and NG/Cu NWs;M50 after DZGP, they were investigated using the FE-SEM. As shown in Figure 2c–f, Cu NWs were physically intact following the DZGP at low temperature. We observed the growth of a graphene shell on Cu NWs.

The presence of graphene, the layer number, and degree of the defect level according to CH_4_ gas flow were investigated through Raman spectroscopy. Figure 3 displays the Raman spectra of the NG/Cu NWs;M20 and NG/Cu NWs;M50 measured at an excitation wavelength of 532 nm. We confirmed the presence of the characteristic Raman spectral peaks of graphene at 1350, 1600, and 2700 cm^−1^ on the NG/Cu NWs;M20 and NG/Cu NWs;M50 samples. The Raman intensities showed the defect-band (D-band) of sp^3^ carbon bond and the graphite-bond (G-band) of the sp^2^ carbon bond. In general, the D-band is caused by the breathing mode of the A1g-symmetry, and involving phonons near the K-zone boundary. However, the G-band is caused by the zone center phonons of the E2g-symmetry [38]. There are many factors, such as doping, layer numbers, defects, strains, and substrate, which can affect the position of the D band. Moreover, our observations are similar to those of N-doped carbon nanotubes (CNTs), in which the degree of N doping causes an upshift of the D band and a downshift of the G band [33,34]. The high intensity of the D peak indicates the doping of the graphitic sheets [33,34], as the D band only occurs in the sp^2^ C with defects [39], and N doping introduces large amounts of topological defects [35,36,37]. To gain a better insight into the G/Cu NWs, the broad Raman spectrum in the range 1100–2000 cm^−1^ was deconvoluted. In addition, four peaks, that is, D1, D2, D3, and D’ were observed. The edge shows a special kind of disorder, and the nature of the edges could be probed with respect to the intensity of the D peak. The D peak arises from the momentum conservation law owing to the formation of edge states (zigzag and armchair). The many studies reported that the intensity of the D peak is very sensitive to the nature of edge states in graphene. The additional peaks D1, D2, and D3 may be associated with functional group defects. The D1 peaks of NG/Cu NWs;M20 and NG/Cu NWs;M50 at 1197 and 1182 cm^−1^ were assigned to the sp^2^–sp^3^ bonded carbon atoms at the graphene edges [40,41,42,43,44]. The D2 peaks of NG/Cu NWs;M20 and NG/Cu NWs;M50 at 1245 and 1248 cm^−1^ were assigned to COOH or ring-type C–OH edge functional groups [40,41,42], while the Raman D3 peaks at 1514 and 1496 cm^−1^ were attributed to C=O/C–O functional groups [42,43]. The D’ peak is attributed to the vacancies and/or pentagonal and octagonal defects, usually referred to as zigzag 5–7 defects, and represents the crystalline defects in graphene [33,42,43,44]. The D’ peaks of NG/Cu NWs;M20 and NG/Cu NWs;M50 were centered at 1607 and 1666 cm^−1^. In addition, the 2D peak appeared when a non-elastic scattering phonon occurred after the second experiment, and the position of the 2D peak was dependent on the number of graphene layers [41,45]. Generally, I_2D_/I_G_ ratio >2 indicates the growth of monolayer graphene, while a value <2 indicates the growth of bilayer graphene. As observed from Figure 3, the ratios of the I_2D_/I_G_ peak intensities were 0.11 and 0.09 for the NG/Cu NWs;M20 and NG/Cu NWs;M50 samples, respectively. Therefore, based on the Raman characterization, the obtained results showed that the NG/Cu NWs;M20 and NG/Cu NWs;M50 samples resulted in N-doped graphene with multilayers on Cu NWs.

The XPS spectra showed the doping of the graphene. Figure 4 displays the high-resolution C 1s, N 1s, O 1s, and Cu 2p_3/2_ spectra of Cu NWs, NG/Cu NWs;M20, and NG/Cu NWs;M50 samples. For as-grown Cu NWs, the C 1s spectra were observed because of the presence of EDA. As shown in Figure 5a, C sp^2^ peaks of all samples were centered at 284.30, 284.53, and 284.53 eV, respectively. The main peak corresponds to the graphite-like C sp^2^, indicating that most of the C atoms in the N-doped graphene were arranged in a conjugated honeycomb lattice. The C sp^2^ peak of as-grown Cu NWs was weaker than the C sp^3^ peak, whereas C sp^2^ peaks of NG/Cu NWs;M20 and NG/Cu NWs;M50 were much stronger than their C sp^3^ peaks. This indicates that most of the carbon atoms in NG/Cu NWs were arranged in a two-dimensional graphite-like honeycomb lattice [28,29]. The C sp^3^ peaks of all samples were located at 284.83, 285.39, and 285.39 eV, respectively. The C sp^3^ peak of as-grown Cu NWs was stronger than the sp^2^ peak, suggesting that the EDA randomly bonded with each other. The C sp^3^ peaks of NG/Cu NWs;M20 had a relatively smaller intensity than those of the NG/Cu NWs;M50. This was attributed to either crystalline or functional-group defects in G/Cu NWs, and corresponds to the N-sp^3^ C bonds. In C 1s spectra of as-grown Cu NWs, the very small peaks at 286.23 and 288.95 eV were attributed to C–O and O–C=O, respectively. For C 1s spectra of NG/Cu NWs;M20 and NG/Cu NWs;M50, the broad peaks were attributed to C–O/C=N, which were located at 285.44 and 285.46 eV, respectively. Other peaks at 288.92 and 289.18 eV were attributed to the C–N functional group. The results indicated that the carbon atoms at the edge are reconstructed during the removal of impurities [46]. In addition, the carbon atoms were bonded with amine groups to maintain the stability of the carbon atoms at the graphene edges. As shown in Figure 5b, we confirmed four peaks in the as-grown Cu NWs, namely, pyridinic N, pyrrolic N, amino N, and oxygenated N, which are located at 398.07, 399.74, 400.70, and 404.14 eV, respectively. After DZGP, the N 1s spectra of NG/Cu NWs;M20 and NG/Cu NWs;M50 also showed four peaks: Pyridinic N, amino N, graphitic N, and oxygenated N. The pyridinic N peaks of NG/Cu NWs;M20 and NG/Cu NWs;M50 were centered at 398.65 and 398.03 eV, respectively. 

The amino-N peaks were notably changed after DZGP. The amino-N peaks of NG/Cu NWs;M20 and NG/Cu NWs;M50 were located at 400.30 and 400.69 eV, respectively. The pyrrolic-N peak disappeared, while the graphitic-N peaks were newly generated. Here, the peaks refer to the N atoms located in a π conjugated system and contribute to the π system with one or two p-electrons, respectively [47,48]. The peak of graphitic N, which refers to the N atoms replacing the C atoms inside of the graphene layers [48], was much higher; thus, the N atoms are substitutionally doped into the graphene lattice and existed mainly in the form of graphitic N. The graphitic-N peaks in NG/Cu NWs;M20 and NG/Cu NWs;M50 were centered at 402.09 and 402.38 eV, respectively. As the amine groups on Cu NWs bonded with the carbon atoms, the amino N and pyrrolic N changed to graphitic N [49,50]. As shown in Figure 4c, the as-grown Cu NWs displayed three peaks, namely, Cu_2_O, Cu(OH)_2_, and C–O, which were located at 530.19, 531.95, and 532.63 eV, respectively. The Cu_2_O peaks in NG/Cu NWs;M20 and NG/Cu NWs;M50 were centered at 530.44 and 530.51 eV, respectively. After the DZGP, the Cu(OH)_2_ peaks were significantly changed, indicating that the Cu NWs were reduced because of the DZGP. In addition, the O 1s peak results from the oxygen or water absorbed on the surface of the N-doped graphene [47,51,52]. As shown in Figure 4d, the Cu 2p_3/2_ spectrum of all samples showed a similar tendency; however, we observed that the peaks were shifted after the DZGP. These results show that we successfully grew the NG/Cu NWs due to the amine groups onto as-grown Cu NWs. Furthermore, the NG/Cu NWs;M50 sample showed a larger amount of N doping than the NG/Cu NWs;M20 sample.

Additionally, the NG/Cu NWs;M20 and NG/Cu NWs;M50 samples were exposed to atmospheric conditions for several days to test their oxidation stability. The EDA with amine group on as-grown Cu NWs was removed using de-ionized (DI) water for observing the oxidation stability of pure Cu NWs. Figure 5a shows the sheet resistances of all samples. The pure Cu NWs were naturally oxidized immediately after the EDA was removed. However, the sheet resistances of NG/CuNWs;M20 and NG/Cu NWs;M50 were retained owing to the N-doped graphene-coated layer. Koo et al. [53] reported on the relationship between the EDA and oxidation of Cu NWs and suggested that the EDA, which is a growth source of N-doped graphene, could be suppressing the oxidation of Cu NWs. Figure 5b results show that the NG/Cu NWs;M50 sample with higher doping than the NG/Cu NWs;M20 sample shows characteristics of low sheet resistance and high oxidation prevention.

The as-grown Cu NWs, NG/Cu NWs;M20, and NG/Cu NWs;M50 were fabricated as the fillers in TIMs at 23 wt% for measuring thermal conductivity through the laser flash method. Table 1 shows the clear values of thermal conductivity and the TIMs sample name about sample number. Figure 6 displays the thermal conductivity of the TIMs. The measured thermal conductivity of the polymer matrix was 0.22 W/m∙K. The thermal conductivities of NG/Cu NWs with M20 and M50 were 0.39 and 0.51 W/m∙K, respectively, showing an improvement by 75% and 130%, respectively. Moreover, the NG/Cu NWs;M50 sample showed good thermal conductivity than that of the previously studied TIMs.

## 4. Conclusions

In this study, we synthesized Cu NWs with diameters of 160–200 nm and lengths of ~25 µm. To avoid the rapid oxidation and changing of the morphology of the Cu NWs, we succeeded in synthesizing NG/Cu NWs according to two CH_4_ gas flow methods by using DZGP at low growth temperature of approximately 600 °C. The morphology of NG/Cu NWs was confirmed to be well preserved. N doping for graphene was successfully conducted through amine groups on as-grown Cu NWs and confirmed by Raman spectra and XPS measurement. The TIMs with three NG/Cu NWs;M20 and NG/Cu NWs;M50 as fillers were prepared using a paste mixer, and their thermal conductivity values were measured using a laser flash method. The thermal conductivity values of the heat-transfer materials (TIMs) with NG/Cu NWs;M20 and NG/Cu NWs;M50 are measured to be 0.39 and 0.51 W/m∙K, respectively, indicating an improvement of 75% and 130%, respectively, compared with the polymer matrix (0.22 W/m∙K). Especially, the NG/Cu NWs;M50 sample showed good thermal conductivity than that of the previously studied TIMs [1,2,3,4,54]. The results showed that the synthesis technology of the N-doped graphene on Cu NWs could promote future research and applications in the TIMs and air-stable flexible electronic devices.

## Figures and Tables

**Figure 1 nanomaterials-09-00984-f001:**
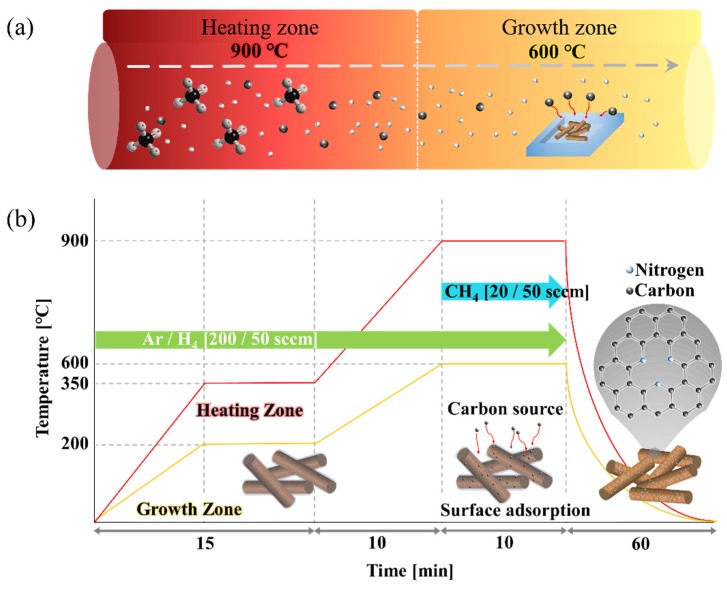
Schematic illustrations for processes in present work: (**a**) double zone growth method and (**b**) detailed method of graphene growth.

**Figure 2 nanomaterials-09-00984-f002:**
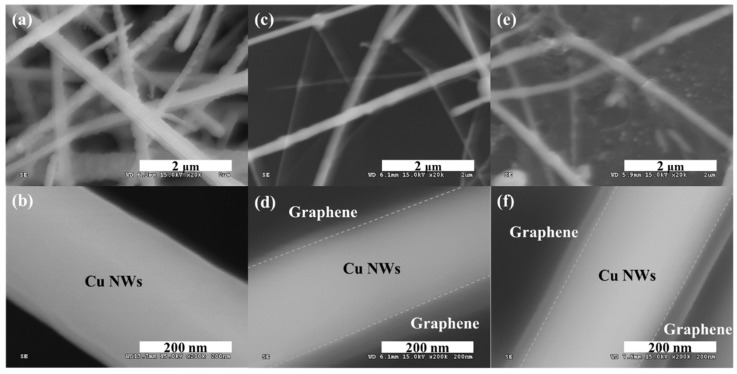
SEM images of (**a**,**b**): As-grown Cu NWs, (**c**,**d**): NG/Cu NWs;M20, and (**e**,**f**): NG/Cu NWs;M50.

**Figure 3 nanomaterials-09-00984-f003:**
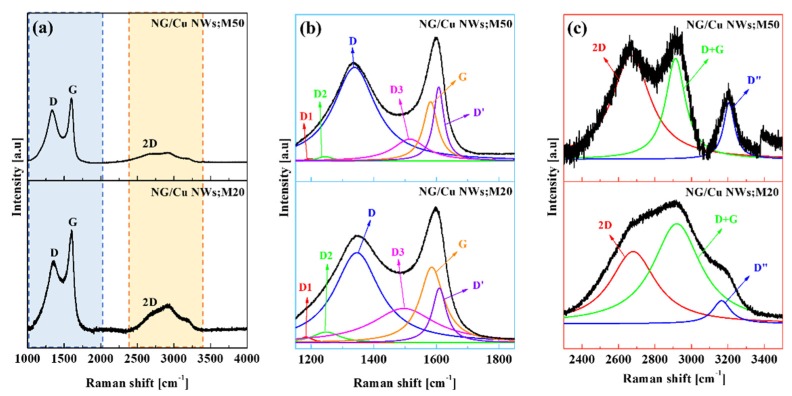
Raman spectra ranging from (**a**) 1000–4000 cm^−1^, (**b**) 1100–1900 cm^−1^, and (**c**) 2300–3500 cm^−1^.

**Figure 4 nanomaterials-09-00984-f004:**
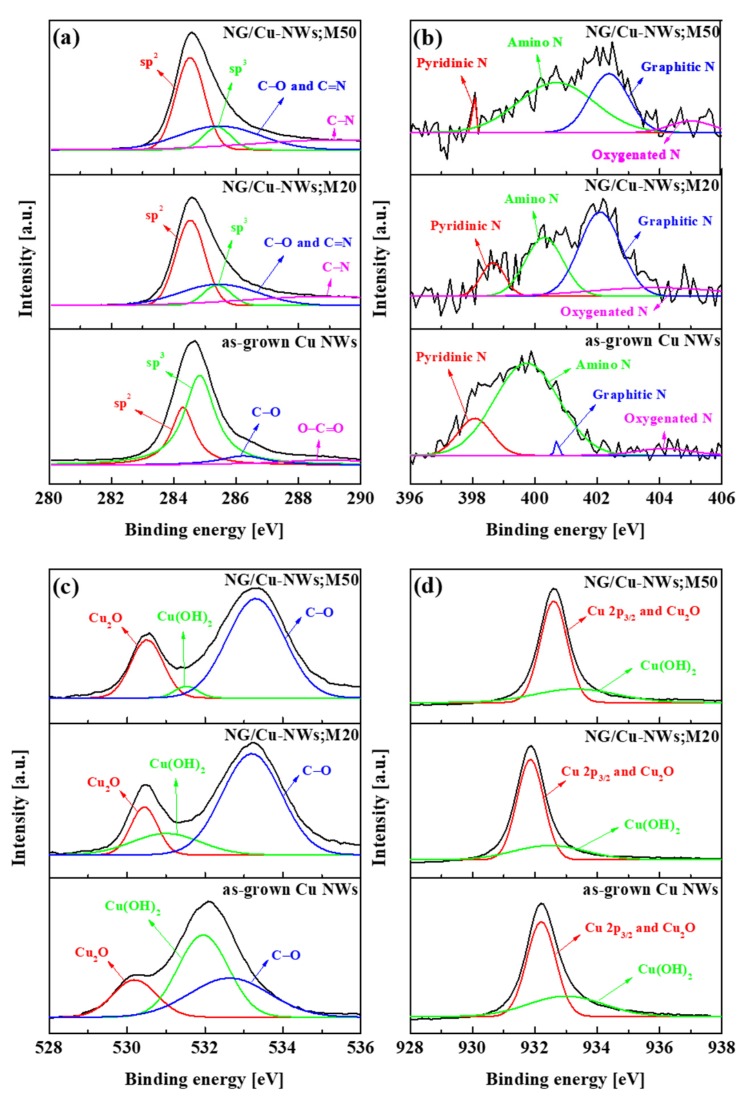
XPS spectra of as-grown Cu NWs, NG/Cu NWs;M20, and NG/Cu NWs;M50: high-resolution (**a**) C 1s, (**b**) N 1s, (**c**) O 1s, and (**d**) Cu 2p_3/2_.

**Figure 5 nanomaterials-09-00984-f005:**
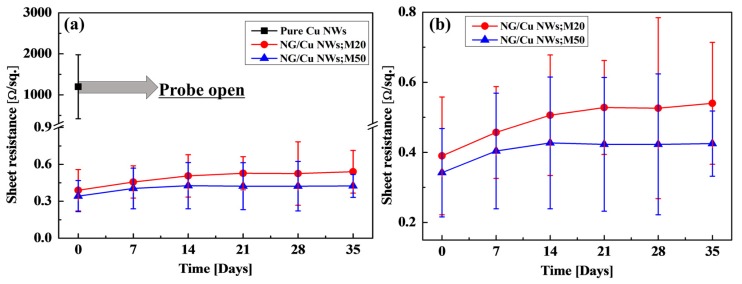
(**a**) Sheet resistance of the pure Cu NWs, NG/Cu NWs;M20, and NG/Cu NWs;M50 and (**b**) the enlarged sheet resistance of NG/Cu NWs;M20 and NG/Cu NWs;M50.

**Figure 6 nanomaterials-09-00984-f006:**
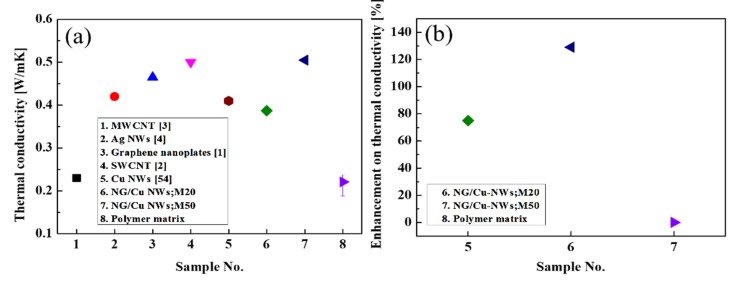
(**a**) Thermal conductivity of the two TIMs used in this study and the TIMs of other studies. (**b**) Enhancement of thermal conductivity of the two TIMs used in the current study from the polymer matrix.

**Table 1 nanomaterials-09-00984-t001:** The thermal conductivity and the TIMs.

Sample No.	TIMs Sample Name	Thermal Conductivity [W/m∙K]	Reference
1	MWCNT	0.23	[3]
2	Ag NWs	0.42	[4]
3	Graphene nanoplates	0.465	[1]
4	SWCNT	0.5	[2]
5	Cu NWs	0.41	[54]
6	NG/Cu NWs;M20	0.387	In this study
7	NG/Cu NWs;M50	0.505	In this study
8	Polymer matrix (Epoxy)	0.221	In this study

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
