# Peer review of "Synthesis of Nitrogen-Doped Graphene on Copper Nanowires for Efficient Thermal Conductivity and Stability by Using Conventional Thermal Chemical Vapor Deposition"

_nanomaterials, 2019, doi:10.3390/nano9070984_

Round 1
Reviewer 1 Report
For the explanation of Raman spectra, more literature is necessary for the justification.
I suggest authors see and cite the following research on C-related Raman spectra: Physical Chemistry Chemical Physics 11 (27), 5628-5633 (2009). Importantly 1-2 sentence is necessary for the origin of defects. Other than this, the manuscript and its results were presented in an outstanding way ant this manuscript should be published after minor revisions. Good work!
Author Response
Thank you for your reviews and suggestions. We revised our manuscript according to reviewers’ comments and suggestions. Then, the manuscript was sent to a professional company for proof reading to check the English grammar and descriptions.

Reviewer 2 Report
I am happy with the paper in its current shape.
Author Response
Thank you very much, i am very happy to your comments.
We clearly checked our manuscript according to reviewers’ comments and suggestions. Then, the manuscript was sent to a professional company for proof reading to check the English grammar and descriptions.
This manuscript is a resubmission of an earlier submission. The following is a list of the peer review reports and author responses from that submission.
Round 1
Reviewer 1 Report
As far as I know defects has nothing to do with G-band in Raman. They are related with D-band. Therefore this sentence should be revised:
"There are many factors, such as doping, layer numbers, defects, strains, and substrate, which can affect the position of the G band." Authors regarding D and G band of carbon materials should see and cite following articles: Scientific reports 7 (1), 11222 (2017); Nanoscale 10 (4), 1877-1884 (2018)
I can tread anything in figure 4. Please enlarge all.
the assignment of D1 D2 D3 and D' is highly speculative and maybe some specific referencing is necessary for each
It is good to give a tabel for sample number. What exactly sample no 1,2,3,4,5,6 is unclear.
The manuscript is really nice and has to be published after some minor revision.
Reviewer 2 Report
In my opinion, this paper cannot be accepted for publication (at least in the present form).
The introduction is kind of messy - authors ''jump'' from one subject to another without a clear explanation of their motivation behind this work.
The figures in the paper are terrible (both in terms of the quality and scientifical content):
Figure 1 - the quality of this image is poor (the figure is blurry and unclear). Figure 1b is not described in the text. The designations for the axis are missing.
Figure 3 - I've got some difficulties in identifying the parts of this image. If I got it correctly, the experimental Raman spectra (obtained for two different samples) are given in black. So, what about the bunch of multicolor peaks that are used to ''decipher'' the experimental data? I assume the positions of these peaks are found through some modeling. Why it is not explained in the text?
Figure 4 (XPS spectra) is a total mess. The image quality is terrible. Again, it seems like the experimental spectra are given in black color. All other peaks (given in different colors) look like modeling results. Why it is not explained in the text?
Figure 5 - the authors claim that: ''These results show that the NG/Cu NWs;M50 sample with higher doping than the NG/Cu NWs;M20 sample shows characteristics of low sheet resistance and high oxidation prevention''. Unfortunately, it does not follow from the Figure. The data obtained for both samples are absolutely identical. Personally, I cannot see any improvements in electrical properties.
Figure 6 makes no physical sense. The thermal conductivity is experimentally measured for two samples (copper nanowires + nitrogen-doped graphene). The results are compared to the measured thermal conductivity of the polymer matrix. This comparison makes no sense. These materials are different - Cu will obviously have better thermal properties. The authors should compare the thermal properties of different Cu nanowires samples (NG/CuNWs vs CuNWs).
By the way, it is not mentioned (in the text) what kind of polymer matrix has been used in the experiment. The information about the substrate for Cu nanowires is also missing.
The authors claim that they studied the morphology of the Cu nanowires samples. I can not see any results on the morphology of Cu nanowires in the paper. A couple of SEM images is not what people usually call the morphology study.
The authors state that they used SEM images to estimate the dimensions of the Cu nanowires. Seems like they estimated them wrongly. Thus, in the paper, it is written that ''The average diameter of the Cu NWs was 160–200 nm and the average length was ~25 μm.'' One can see (for example, from Figs 2b, 2d, 2f ) that the diameter is close to 400 nm.
Reviewer 3 Report
Overall I found the paper interesting and rather complete. There are a number of minor points to be assessed. First of all I suggest a revision of English from a mother tongue.
Abstract
line 14 "in modern"
line 16 to 18 rephrase, unclear
line 22 rephrase, unclear
line 26 "as heat transfer"
Introduction
line 34 "non-efficient"
line 41 cite relevant literature [Composites: Part A 61 (2014) 108–114]
line 60/62 please better clarify why substitutional
Materials & Methods
line 92 "anisotropic"
Results & discussion
line 173 why double 284.53?
line 179 why double 285.39?
Figure 6 y axis W/mK upper case K; panel (b) I think the dots in the legend are not corresponding to what we see in the figure